# Adaptive Sliding Mode Fault Compensation for Sensor Faults of Variable Structure Hypersonic Vehicle

**DOI:** 10.3390/s22041523

**Published:** 2022-02-16

**Authors:** Kai-Yu Hu, Chunxia Yang, Wenjing Sun

**Affiliations:** 1Aerospace Software Evaluation Center, Beijing Jinghang Institute of Computing and Communication, Beijing 100074, China; zian_cheng@126.com (C.Y.); ecywen@126.com (W.S.); 2Applied Mathematics Research Center, China Aerospace Science and Industry Corporation, Beijing 100074, China; 3College of Automation, Nanjing University of Aeronautics and Astronautics, Nanjing 211106, China

**Keywords:** hypersonic flight vehicle, sensor faults, fault-tolerant control, adaptive dynamic surface, sliding mode method

## Abstract

This paper investigates the sensor fault detection and fault-tolerant control (FTC) technology of a variable-structure hypersonic flight vehicle (HFV). First, an HFV nonlinear system considering sensor compound faults, disturbance, and the variable structure parameter is established, which is divided into the attitude angle outer and angular rate inner loops. Then a nonlinear fault integrated detector is proposed to detect the moment of fault occurrence and provide the residual to design the sliding mode equations. Furthermore, the sliding mode method combined with the virtual adaptive controller constitutes the outer loop FTC scheme, and the adaptive dynamic surface combined with the disturbance estimation constitutes the inner loop robust controller; these controllers finally realize the direct compensation of the compound sensor faults under the disturbance condition. This scheme does not require fault isolation and diagnosis observer loops; it only uses a variable structure FTC with a direct estimation algorithm and integrated residual to complete the self-repairing stable flight of variable-structure HFV, which exhibits a high reliability and quick response. Lyapunov theory proved the stability of the system, and numerical simulation proved the effectiveness of the FTC scheme.

## 1. Introduction

The attitude control technology is an important research direction for hypersonic flight vehicles (HFVs) [1,2,3]. The reentry HFV is in a complex flight environment, with a flying speed of up to Mach 30 [4,5]. Pneumatic heating at high speeds makes the attitude system a nonlinear system with a strong disturbance, long time delay, and easy failure [6,7,8]. We design a fault-tolerant controller for the reentry attitude system with variable-structure parameter to solve the problem of sensor fault self-healing; thus, the malfunctioning HFV can stably track the super maneuvering reference command and quickly return to normal flight.

To study the fault-tolerant control (FTC) of HFVs, control problems, such as nonlinearity, disturbance, and parameter uncertainty under nominal flight conditions, must be solved. Nonlinearity is an inherent property of HFV; in [9], an indirect global neural controller of strict-feedback systems in the presence of unknown nonlinear dynamics was designed using the dynamic surface. In [10], the proposed control method for the nonlinear disturbance observer solved the problem of robust flight control of the aspirating HFV under a mismatch disturbance. Disturbance is the primary problem that HFVs face in harsh environments [11,12]. In [13], the robustness of the proposed HFV controller was enhanced using a disturbance observer, which eliminated the detrimental couplings while maintaining the beneficial couplings. The presented controller in [14] was completely appropriate for HFVs with disturbances; however, it did not rely on an accurate model. Indeed, it was data-driven and could adjust its parameters online under various operating conditions. In [15], an optimal fuzzy approximation strategy was designed to reproduce the uncertain functions for compound uncertainties composed of nonlinear uncertain parameters and environmental disturbances. In [16], the proposed multi-variable integral sliding-mode control could guarantee the finite-time stability in the presence of actuator malfunctions and disturbances. However, these controllers were all designed under fault-free conditions; thus, we employed the adaptive disturbance estimation combined with the direct compensation of the nonlinear dynamic surface to suppress the disturbance in the inner loop of HFVs, and then combined the adaptive sliding mode and dynamic surface to achieve a nonlinear fault compensation.

Furthermore, HFV is prone to failure owing to its complex structure and harsh environment, and, thus, actuator faults have been significantly studied recently [17,18]. In [19], a fuzzy reinforcement learning-based tracking control algorithm was first proposed for partially unknown systems with actuator faults. In [20], a compound control scheme combining the fractional-order proportional-integral-derivative and linear active disturbance rejection controls was investigated for reentry HFVs with actuator faults. In [21], the adaptive compensation control law was presented considering both the unknown uncertainty and unexpected actuator faults, including additive and multiplicative faults. In [22], the adaptive technique was combined with the sliding mode FTC design to guarantee the asymptotical convergence of tracking errors and deal with unknown actuator faults. However, the above methods did not consider sensor faults caused by pneumatic heating; this paper will solve the problem of sensor fault self-repair.

The model-based fault diagnosis methods have been employed in investigating HFVs, satellites, or other spacecraft to consider sensor faults, where the core concept is to design observers for fault diagnosis [23,24,25]. In [26], a fault diagnosis scheme based on the data-driven observer of HFV was introduced, the sensor fault features were obtained using wavelet translation, and a distance evaluation technique based on the Spearman correlation analysis selected features. In [27], the observer comprehensively diagnosed sensor faults according to the fault level monitored by a sequential probability ratio test. In [28], an adaptive control scheme was proposed for HFVs with disturbance, input saturation, and detection sensor faults of states. In [29], the longitudinal HFV was transformed into velocity and altitude subsystems, considering the full state sensor constraints. In addition, the finite-time stability-based controller with a robust adaptive distributive law was constructed to deal with the observed sensor faults. However, designing observers for a nonlinear variable-structure HFV system is increasingly difficult because additional fault observers increase the system complexity. Moreover, a larger number of observers with higher dimensions are required to deal with compound faults, resulting in greater challenges to the system stability.

In this study, an FTC scheme is developed using the indirect variable structure adaptive, sliding mode, and dynamic surface methods for a nonlinear HFV with sensor compound faults and disturbances. This study achieves a simple robust compensation strategy without independent fault isolation and diagnostic control loops, and improves the reliability of super maneuverable flight with a variable structure. Therefore, the main contributions of this study are as follows:On the basis of the first established variable structure system of reentry HFV with sensor compound faults and disturbance, a nonlinear fault detector is designed to generate weighted integrated residual to detect sensor faults;The original system is divided into outer and inner loops, an adaptive fault-tolerant virtual controller and a dynamic surface controller are designed, respectively, to ensure the stability of the system under sensor compound faults in the outer loop and disturbance in the inner loop;The weighted integrated residual and adaptive compensation item is added to the controller to directly compensate for sensor faults, so the faults can be repaired without fault isolation and diagnosis observer loops, which simplifies the algorithm and shortens the response time.

The main content of this paper is: Section 2 presents the reentry attitude model of variable-structure HFV with sensor compound faults and disturbance, as well as the FTC target; Section 3 presents the design method of nonlinear fault detector, and the overall FTC scheme including the adaptive fault-tolerant virtual controller in outer loop and the robust dynamic surface controller in inner loop; Section 4 shows a comparative simulation experiment using nominal and fault-tolerant controller for the variable-structure HFV with sensor faults. Section 5 summarizes the full paper.

## 2. Variable-Structure HFV Model with Sensor Faults

First, the reentry HFV system with the variable structure parameter, sensor compound faults, and disturbances is introduced. We consider the six-degree-of-freedom attitude kinematics model of X-33 and X-38, which was introduced in [30]. The model is composed of three attitude angle and three angular rate equations. When the rotation of the earth is ignored, the model is simplified as follows:(1)$$\left\{ \begin{array}{l} \dot{\theta }=R\omega \\ \dot{\omega }=-{\left(J+J_{VS}\right)}^{-1}\Omega (J+J_{VS})\omega +{\left(J+J_{VS}\right)}^{-1}T+d(t)\\ y=\theta +f_{\theta }(t) \end{array}\right.$$
where *ω* = [*p*, *q*, *r*]*^T^* is the measurable angular rate vector, including the roll angular rate *p*, the pitch angular rate *q*, and the yaw angular rate *r*. *θ* = [*ϕ*, *α*, *β*]*^T^* is the attitude angle vector, including the roll angle *ϕ*, the attack angle *α*, and the angle of sideslip *β*. *J* ∈ R^3×3^ is a symmetric positive matrix representing the inertia. *J_VS_* ∈ R^3×3^ is the inertia perturbation caused by the active deformation of the fuselage; that is, the variable structure parameter, which represents the physical phenomenon of a HFV changing the fuselage structure. Moreover, *T* ∈ R^3×1^ is the control input torque, *d*(*t*) ∈ R^3×1^ represents an external disturbance, and *f_θ_*(*t*) ∈ R^3×1^ is the compound sensor faults of the output angle channels. The control input torque *T* can be expressed as follows:(2)T=Ψu
where *u* = [*δ_e_*, *δ_a_*, *δ_r_*]*^T^* is the control input vector, including the elevator deflection angle *δ_e_*, aileron deflection angle *δ_a_*, and rudder deflection angle *δ_r_*. Some system parameter matrices are as follows:(3)$$\Psi =\left[\begin{array}{ccc} g_{p,\delta_{e}} & g_{p,\delta_{a}} & g_{p,\delta_{r}}\\ g_{q,\delta_{e}} & g_{q,\delta_{a}} & g_{q,\delta_{r}}\\ g_{r,\delta_{e}} & g_{r,\delta_{a}} & g_{r,\delta_{r}} \end{array}\right],\;R=\left[\begin{array}{ccc} -\cos\alpha \cos\beta & -\sin\beta & -\sin\alpha \cos\beta \\ -\cos\alpha \tan\beta & 1 & -\sin\alpha \tan\beta \\ \sin\alpha & 0 & -\cos\alpha \end{array}\right],\;\Omega =\left[\begin{array}{ccc} 0 & -r & q\\ r & 0 & -p\\ -q & p & 0 \end{array}\right]$$Ψ ∈ R^3×3^ is the rudder allocation matrix, the elements of the matrix are the allocation weights of the three attitude control signals in the three attitude control loops, which was thoroughly described in [30].

**Assumption** **1.**
*The disturbance is continuous and bounded; that is, there are two positive constants d_1b_ and d_2b_ that make (4) hold. The derivative of disturbance is derivable.*



(4)
$$\left\{ \begin{array}{l} \left\|\dot{d}\right\|\leq d_{2b}\\ \left\|d\right\|\leq d_{1b} \end{array}\right.$$


**Assumption** **2.**
*Sensor compound faults are continuous and bounded; that is, there are two positive constants f_1b_ and f_2b_ that make (5) hold. The derivative of sensor compound faults is derivable.*



(5)
$$\left\{ \begin{array}{l} \left\|{\dot{f}}_{\theta }\right\|\leq f_{2b}\\ \left\|f_{\theta }\right\|\leq f_{1b} \end{array}\right.$$


The goal of this study is to design an adaptive FTC scheme for the reentry attitude model of a variable-structure HFV with sensor compound faults and disturbances, which makes the closed-loop system state and output bounded, and the attitude angle *θ* can accurately track the reference command *θ_d_*. A control block diagram is shown in Figure 1.

## 3. Fault Integrated Detection and FTC Design

Based on the HFV model with sensor compound faults in Section 2, this section first designs a nonlinear observer to detect faults, then judges whether to add the fault compensation item. Subsequently, a fault-tolerant controller based on the adaptive sliding mode, dynamic surface, and variable structure harmonic functions is designed. A sliding surface is designed according to the reference outputs of the attitude angles to obtain the virtual control variable *ω_d_* transmitted to the inner angular rate loop. Simultaneously, the virtual adaptive controller directly compensates for the sensor faults such that the three output attitude angles with faults track the reference command again. This method can directly compensate for faults without fault isolation or diagnosis. Finally, the virtual FTC is fed forward into the inner loop dynamic surface controller to obtain the required control input torque *T*.

### 3.1. Design of Nonlinear Fault Detector

The noninterference situation is considered to simplify the design process of the detection observer based on the model expressed in (1). When the state vector is defined as *x* = [*θ^T^ ω^T^*]*^T^*, then the HFV reentry attitude model can be described as
(6)$$\left\{ \begin{array}{l} \dot{x}(t)=\phi (x)x(t)+B^{*}T\\ y_{d}(t)=Cx(t)+f(t) \end{array}\right.$$
where
(7)$$\phi (x)=\left[\begin{array}{cc} 0_{3} & R\\ 0_{3} & -{\left(J+J_{VS}\right)}^{-1}\Omega (J+J_{VS}) \end{array}\right]$$
(8)$$\begin{array}{l} B^{*}=\left[\begin{array}{c} 0_{3}\\ {\left(J+J_{VS}\right)}^{-1} \end{array}\right]\\ =\left[\begin{array}{c} 0_{3}\\ J^{-1} \end{array}\right]+\Delta B\\ =B+\Delta B \end{array}$$
(9)$$C=I_{6}$$
(10)$$f(t)={\left[\begin{array}{cc} f_{\theta }^{T} & 0 \end{array}\right]}^{T}$$

Design the following nonlinear fault detection observer for the new system (6):(11)$$\left\{ \begin{array}{l} \dot{\hat{x}}(t)=\psi (\hat{x},y_{d})+ϒ(y_{d},\alpha^{*},\beta^{*})(y_{d}(t)-{\hat{y}}_{d}(t))+(B+\Delta B)T\\ {\hat{y}}_{d}(t)=C\hat{x}(t) \end{array}\right.$$
where $\hat{x}$ is the state estimation,${\hat{y}}_{d}={\left[\begin{array}{cc} {\hat{\theta }}^{T} & {\hat{\omega }}^{T} \end{array}\right]}^{T}$ is the estimated output. Equations (12) and (13) are the gain matrices of the detection observer and *α*^*^ > 0, *β*^*^ > 0.
(12)$$\psi (\hat{x},y_{d})=\left[\begin{array}{c} R\hat{\omega }\\ -{\left(J+J_{VS}\right)}^{-1}\Omega (\hat{\omega })(J+J_{VS})\omega \end{array}\right]$$
(13)$$ϒ(y_{d},\alpha^{*},\beta^{*})=\left[\begin{array}{cc} \alpha^{*}I & 0_{3}\\ {\left(J+J_{VS}\right)}^{-1}R^{T} & \beta^{*}{\left(J+J_{VS}\right)}^{-1} \end{array}\right]$$

Define the output error as follows:(14)$${\tilde{y}}_{d}(t)=y_{d}(t)-{\hat{y}}_{d}(t)$$
(15)$${\tilde{y}}_{d}(t)={\left[\begin{array}{cc} \xi_{\theta }^{T} & \xi_{\omega }^{T} \end{array}\right]}^{T}$$
where
(16)$$\xi_{\theta }=\theta -\hat{\theta }$$
(17)$$\xi_{\omega }=\omega -\hat{\omega }.$$

Subsequently, the nonlinear fault detector described by Equation (11) can be expressed as
(18)$$\left\{ \begin{array}{l} \dot{\hat{\theta }}=R\hat{\omega }+\alpha^{*}\xi_{\theta }\\ \dot{\hat{\omega }}=-{\left(J+J_{VS}\right)}^{-1}\Omega (\hat{\omega })(J+J_{VS})\hat{\omega }+{\left(J+J_{VS}\right)}^{-1}\beta^{*}\xi_{\omega }\\ +{\left(J+J_{VS}\right)}^{-1}R^{T}\xi_{\theta }+{\left(J+J_{VS}\right)}^{-1}T \end{array}\right.$$

**Theorem** **1.**
*When the sensor has no fault, that is, f(t) = 0, by selecting appropriate gains of the nonlinear observer α^*^ > 0 and β^*^ > 0, the output estimation error (15) converges asymptotically.*


**Proof of Theorem** **1.**Define the following Lyapunov function:(19)$$V_{\xi }=\frac{1}{2}{\tilde{y}}_{d}^{T}P{\tilde{y}}_{d}$$
where



(20)
$$P=\left[\begin{array}{cc} I & 0_{3}\\ 0_{3} & J+J_{VS} \end{array}\right]$$



The derivative of formula (19) with respect to time is:(21)$$\begin{array}{l} {\dot{V}}_{\xi }=\xi_{\theta }^{T}{\dot{\xi }}_{\theta }+\xi_{\omega }^{T}(J+J_{VS}){\dot{\xi }}_{\omega }\\ =\xi_{\theta }^{T}(\dot{\theta }-\dot{\hat{\theta }})+\xi_{\omega }^{T}(J+J_{VS})(\dot{\omega }-\dot{\hat{\omega }}) \end{array}$$

Ignore the two minimal error multiplication of the second term in (21), then consider (1) and (18), the above formula can be rewritten as
(22)$$\begin{array}{l} {\dot{V}}_{\xi }=\xi_{\theta }^{T}(R\omega -R\hat{\omega }-\alpha^{*}\xi_{\theta })+\xi_{\omega }^{T}[-\Omega (\omega )(J+J_{VS})\omega +T\\ +\Omega (\hat{\omega })(J+J_{VS})\omega -\beta^{*}\xi_{\omega }-R^{T}\xi_{\theta }-T]\\ =-\alpha^{*}{\left\|\xi_{\theta }\right\|}^{2}-\beta^{*}{\left\|\xi_{\omega }\right\|}^{2} \end{array}$$

Therefore, if the appropriate gains *α*^*^ > 0 and *β*^*^ > 0 are selected, the sensor output estimation error will gradually converge to zero without fault. □

We design norm (23) as the weighted integrated detection residual:(23)$$r^{*}=\left\|W{\tilde{y}}_{d}(t)\right\|=\left\|diag(w_{1},\ldots ,w_{6}){\tilde{y}}_{d}(t)\right\|$$
where *w_i_* ∈ R (*i* = 1,...,6) is the preset constant weight. The following fault detection mechanism is designed: when *r*^*^ < *T_d_*, no sensor fault occurs, and when *r*^*^ ≥ *T_d_*, sensor fault occurs. *T_d_* is the fault detection threshold, which can be expressed in the following form:(24)$$T_{d}=\sigma T_{d0}+\chi$$
where *T_d_*_0_ is a predetermined threshold when the disturbance is not considered. In addition, *σ* and *χ* are two adjustable positive constants depending on the magnitude of the disturbance and variable structure parameter.

**Remark** **1.**
*Adjusting all w_i_ can prevent false negative or false reports, and setting a single w_i_ improves the detection capability of a specific loop. Replacing distributed residuals with weighted integrated residuals can prevent multiple residuals from disturbing the controller’s judgment. The common sensor fault range of the HFV is fixed. Through repeated experiments, W can be particularly designed to directly repair any fault circuit without isolation or diagnosis.*


When the fault detector detects the faults of the variable-structure HFV model, an FTC scheme needs to be designed to guarantee that the attitude angle output still stably tracks the reference signal in the presence of the sensor faults.

### 3.2. Virtual FTC with Adaptive Sliding Mode in Outer Loop

This section designs a virtual fault-tolerant controller with an anti-saturation function to compensate for complex compound faults. When sensor faults occur, the faulty system is divided into an outer loop containing sensor faults (Equation (25)) and an inner loop containing disturbance (Equation (26)), namely:(25)$$\dot{\theta }=R\omega -{\dot{f}}_{\theta }$$
(26)$$\dot{\omega }=-{\left(J+J_{VS}\right)}^{-1}\Omega (J+J_{VS})\omega +{\left(J+J_{VS}\right)}^{-1}T+d(t)$$

First, for the outer loop, by referring to [30] the following variable structure sliding mode surface is designed:(27)$$\left\{ \begin{array}{l} S=\tilde{\theta }+\int_{0}^{t}[(K_{1}+K_{1}^{*}(J_{VS})){\left|\tilde{\theta }\right|}^{\eta_{1}}sgn(\tilde{\theta })+(K_{2}\\ +K_{2}^{*}(J_{VS})){\left|\tilde{\theta }\right|}^{\eta_{2}}sgn(\tilde{\theta })]d\tau \\ \tilde{\theta }=\theta -\theta_{d} \end{array}\right.$$
where *η*_1_ ≥ 1, 0 < *η*_2_ < 1, and *K_j_* and *K_j_*^*^(*J_VS_*) satisfy:(28)$$\left\{ \begin{array}{l} K_{j}=diag\{k_{j}{}^{1},k_{j}{}^{2},k_{j}{}^{3}\}\\ K_{j}{}^{*}(J_{VS})=diag\{k_{j}{{}^{1}}^{*}(||J_{VS}||),k_{j}{{}^{2}}^{*}(||J_{VS}||),k_{j}{{}^{3}}^{*}(||J_{VS}||)\} \end{array}\right.,j=1,2$$
where *K_j_* is the diagonal function matrix for the fixed-structure flight mode, *K_j_*^*^(*J_VS_*) is the harmonic function matrix to deal with the variable structure flight mode, and *θ_d_* is the reference command for the attitude angles. The angle feedback error satisfies:(29)$$\left|\tilde{\theta }\right|=diag\{\left|{\tilde{\theta }}_{1}\right|,\left|{\tilde{\theta }}_{2}\right|,\left|{\tilde{\theta }}_{3}\right|\}$$

The derivative of the sliding mode surface (27), can be obtained as follows:(30)$$\begin{array}{l} \dot{S}=\dot{\tilde{\theta }}+(K_{1}+K_{1}^{*}(J_{VS})){\left|\tilde{\theta }\right|}^{\eta_{1}}\mathrm{sgn}(\tilde{\theta })+(K_{2}\\ +K_{2}^{*}(J_{VS})){\left|\tilde{\theta }\right|}^{\eta_{2}}\mathrm{sgn}(\tilde{\theta })\\ =R\omega_{c}-{\dot{f}}_{\theta }-{\dot{\theta }}_{d}+(K_{1}+K_{1}^{*}(J_{VS})){\left|\tilde{\theta }\right|}^{\eta_{1}}\mathrm{sgn}(\tilde{\theta })\\ +(K_{2}+K_{2}^{*}(J_{VS})){\left|\tilde{\theta }\right|}^{\eta_{2}}\mathrm{sgn}(\tilde{\theta }) \end{array}$$

To enable the sliding mode system (30) to gradually reach the sliding surface, the variable structure arrival rate is designed as follows:(31)$$\dot{S}=-(\varepsilon_{1}+\varepsilon_{1}^{*}(||J_{VS}||))S-(\varepsilon_{2}+\varepsilon_{2}^{*}(||J_{VS}||)){\left|S\right|}^{\eta_{3}}\mathrm{sgn}(S)$$
where *ε*_1_ and *ε*_2_ are positive scalars, *ε*_1_^*^ and *ε*_2_^*^ are the variable structure harmonic functions, and *η*_3_ satisfies 0 < *η*_3_ < 1. By substituting (31) into (30), the virtual controller with the adaptive sliding mode and sensor faults is obtained as follows:(32)$$\begin{array}{l} \omega_{c}=R^{-1}[{\dot{\theta }}_{d}-(K_{1}+K_{1}^{*}(J_{VS})){\left|\tilde{\theta }\right|}^{\eta_{1}}sgn(\tilde{\theta })-(K_{2}\\ +K_{2}^{*}(J_{VS})){\left|\tilde{\theta }\right|}^{\eta_{2}}sgn(\tilde{\theta })-(\varepsilon_{1}+\varepsilon_{1}^{*}(||J_{VS}||))S-(\varepsilon_{2}\\ +\varepsilon_{2}^{*}(||J_{VS}||)){\left|S\right|}^{\eta_{3}}sgn(S)+{\dot{\hat{f}}}_{\theta }] \end{array}$$
where the last item is the estimated value of the fault derivative ${\dot{f}}_{\theta }$, which can directly compensate for the faults. All freely set parameters and functions in (32) are smoothed. According to the integrated detection in Section 3.1, when no fault exists, the last item is omitted, other items remain unchanged to form a nominal controller; when a fault is detected, the item ${\dot{\hat{f}}}_{\theta }$ is added to fix the fault. The adaptive law is designed as
(33)$${\hat{f}}_{\theta }=-\varphi S$$
where ${\hat{f}}_{\theta }$ is the estimated value of sensor fault *f_θ_*, *φ* = *diag*{*γ*_1_, *γ*_2_, *γ*_3_} is the adaptive gain matrix, and define:(34)$$\left\{ \begin{array}{l} {\tilde{f}}_{\theta }={\hat{f}}_{\theta }-f_{\theta }\\ {\dot{\tilde{f}}}_{\theta }={\dot{\hat{f}}}_{\theta }-{\dot{f}}_{\theta } \end{array}\right.$$
where ${\tilde{f}}_{\theta }$ is the fault estimation error, ${\dot{\tilde{f}}}_{\theta }$ is the derivative of the error.

The discontinuous point derivation of the sign function adopts the method of subtracting adjacent sampling points and dividing by the sampling time. Inequality (62) can be replaced by direct derivation, avoiding the non-derivable problem in the proof.

**Remark** **2.**
*The proposed adaptive sliding mode controller can quickly deal with the effect of vibrations caused by sensor faults by adding an adaptive fault tolerance item. η_2_ and η_3_ are linear bounded functions with the integrated residual r^*^ as the independent variable. The relative and absolute weight change of a certain loop in r^*^ can bypass fault isolation and diagnosis, respectively, to realize the direct integrated FTC, which helps simplify the complex variable structure system.*


**Theorem** **2.**
*For the outer attitude angle loop (25) with sensor compound faults, the adaptive sliding mode virtual FTC (32) can cause the sliding mode surface S to converge to the bounded interval, and the attitude angle θ can gradually track the provided reference command θ_d_.*


**Proof of Theorem** **2.**Define the following Lyapunov function including the sliding mode surface *S* and the fault estimation error ${\tilde{f}}_{\theta }$:(35)$$V_{S}=\frac{1}{2}S^{T}S+\frac{1}{2}{\tilde{f}}_{\theta }^{T}\varphi^{-1}{\tilde{f}}_{\theta }$$

Thence the derivation can be obtained:(36)$$\begin{array}{l} {\dot{V}}_{S}=S^{T}\dot{S}+{\tilde{f}}_{\theta }^{T}\varphi^{-1}\dot{\tilde{f}}\\ =S^{T}[\dot{\tilde{\theta }}+(K_{1}+K_{1}^{*}(J_{VS})){\left|\tilde{\theta }\right|}^{\eta_{1}}\mathrm{sgn}(\tilde{\theta })+(K_{2}\\ +K_{2}^{*}(J_{VS})){\left|\tilde{\theta }\right|}^{\eta_{2}}\mathrm{sgn}(\tilde{\theta })]+{\tilde{f}}_{\theta }^{T}\varphi^{-1}\dot{\tilde{f}}\\ =S^{T}[R\omega_{c}-{\dot{f}}_{\theta }-{\dot{\theta }}_{d}+(K_{1}+K_{1}^{*}(J_{VS})){\left|\tilde{\theta }\right|}^{\eta_{1}}\mathrm{sgn}(\tilde{\theta })\\ +(K_{2}+K_{2}^{*}(J_{VS})){\left|\tilde{\theta }\right|}^{\eta_{2}}\mathrm{sgn}(\tilde{\theta })]+{\tilde{f}}_{\theta }^{T}\varphi^{-1}\dot{\tilde{f}} \end{array}$$

Substituting the virtual fault-tolerant controller (32) into Equation (36), we can obtain:(37)$$\begin{array}{l} {\dot{V}}_{S}=-S^{T}[(\varepsilon_{1}+\varepsilon_{1}^{*}(||J_{VS}||))S+(\varepsilon_{2}+\varepsilon_{2}^{*}(||J_{VS}||)){\left|S\right|}^{\eta_{3}}\mathrm{sgn}(S)-{\dot{\tilde{f}}}_{\theta }]\\ +{\tilde{f}}_{\theta }^{T}\varphi^{-1}{\dot{\tilde{f}}}_{\theta }\\ =-S^{T}[(\varepsilon_{1}+\varepsilon_{1}^{*}(||J_{VS}||))S+(\varepsilon_{2}+\varepsilon_{2}^{*}(||J_{VS}||)){\left|S\right|}^{\eta_{3}}\mathrm{sgn}(S)-{\dot{\tilde{f}}}_{\theta }]+({\hat{f}}_{\theta }^{T}\\ -f_{\theta }^{T})\varphi^{-1}{\dot{\tilde{f}}}_{\theta }\\ =-(\varepsilon_{1}+\varepsilon_{1}^{*}(||J_{VS}||)){\left\|S\right\|}^{2}-(\varepsilon_{2}+\varepsilon_{2}^{*}(||J_{VS}||)){\left\|S\right\|}^{\eta_{3}+1}-f_{\theta }^{T}\varphi^{-1}{\dot{\tilde{f}}}_{\theta }\\ \leq -(\varepsilon_{1}+\varepsilon_{1}^{*}(||J_{VS}||)){\left\|S\right\|}^{2}-(\varepsilon_{2}+\varepsilon_{2}^{*}(||J_{VS}||)){\left\|S\right\|}^{\eta_{3}+1}+\frac{1}{\lambda_{\min}(\varphi )}\left\|f_{\theta }^{T}{\dot{\tilde{f}}}_{\theta }\right\|\\ \leq -(\varepsilon_{1}+\varepsilon_{1}^{*}(||J_{VS}||)){\left\|S\right\|}^{2}-(\varepsilon_{2}+\varepsilon_{2}^{*}(||J_{VS}||)){\left\|S\right\|}^{\eta_{3}+1}+\frac{\kappa }{\lambda_{\min}(\varphi )}\\ \leq -(\varepsilon_{1}+\varepsilon_{1}^{*}(||\Delta J||)){\left\|S\right\|}^{2}-(\varepsilon_{2}+\varepsilon_{2}^{*}(||\Delta J||)){\left\|S\right\|}^{\eta_{3}+1}+\frac{\kappa }{\lambda_{\min}(\Lambda )} \end{array}$$
where
(38)$$\kappa =\left\|f_{\theta }^{T}\dot{\tilde{f}}\right\|$$*λ_min_*(*φ*) represents the smallest eigenvalue of the matrix *φ*, therefore the system will converge to the following region:(39)$$\left\|S\right\|\leq \min\left\{{\left(\frac{\kappa }{\left(\varepsilon_{1}+\varepsilon_{1}^{*}(J_{VS}))\lambda_{\min}(\varphi \right)}\right)}^{1/2},{\left(\frac{\kappa }{\left(\varepsilon_{2}+\varepsilon_{2}^{*}(J_{VS}))\lambda_{\min}(\varphi \right)}\right)}^{1/(\eta_{3}+1)}\right\}$$

By selecting appropriate parameters *ε*_1_, *ε*_2_, *ε*_1_^*^, ε_2_^*^, and *φ*, the system can converge to the ideal interval; that is, under the designed adaptive variable structure controller, the attitude angles gradually track the reference commands under fault condition. The design of Λ will be given in Section 3.3, it does not affect the outer loop stability. □

**Remark** **3.***Compared with the method used for sensor faults in* [31]*, the FTC method in this study directly provides an adaptive fault-learning law in the controller, instead of simply using a constant value greater than the failure norm to compensate.*
*This makes the proposed FTC method highly accurate in dealing with compound faults.*

### 3.3. Adaptive Dynamic Surface Controller in Inner Loop

After the adaptive virtual control *ω_c_* is obtained, the signal is introduced into the following first-order filter to obtain the desired angular rate, and the required control torque is subsequently calculated by the dynamic surface controller to stabilize the inner angular rate loop. Moreover, using a simple dynamic surface in the inner loop can simplify the design of the overall controller and avoid sliding mode chattering. The first-order filter is expressed as
(40)$$\left\{ \begin{array}{l} \tau {\dot{\omega }}_{d}+\omega_{d}=\omega_{c}\\ \omega_{c}(0)=\omega_{d}(0) \end{array}\right.$$
where *τ* > 0 represents the filter time constant. Substituting it into the outer loop expression (25), the attitude angle tracking error can be derived as follows:(41)$$\begin{array}{l} \dot{\tilde{\theta }}=\dot{\theta }-{\dot{\theta }}_{d}=R(\omega -\omega_{d}+\omega_{d}-\omega_{c}+\omega_{c})-{\dot{f}}_{\theta }-{\dot{\theta }}_{d}\\ =R(\tilde{\omega }+\bar{\mu }+\omega_{c})-{\dot{f}}_{\theta }-{\dot{\theta }}_{d}\\ =-(K_{1}+K_{1}^{*}(||J_{VS}||)){\left|\tilde{\theta }\right|}^{\eta_{1}}\mathrm{sgn}(\tilde{\theta })-(K_{2}\\ +K_{2}^{*}(||J_{VS}||)){\left|\tilde{\theta }\right|}^{\eta_{2}}\mathrm{sgn}(\tilde{\theta })-(\varepsilon_{1}+\varepsilon_{1}^{*}(||J_{VS}||))S\\ -(\varepsilon_{2}+\varepsilon_{2}^{*}(||J_{VS}||)){\left|S\right|}^{\eta_{3}}\mathrm{sgn}(S)+\dot{\tilde{f}}+R\tilde{\omega }+R\bar{\mu } \end{array}$$
where
(42)$$\bar{\mu }=\omega_{d}-\omega_{c}$$
(43)$$\tilde{\omega }=\omega -\omega_{d}$$Moreover, (42) represents the filtering error and (43) is the angular rate tracking error. According to the inner loop angular rate system (26) with disturbance, the angular rate error can be expressed as follows:(44)$$\begin{array}{l} \dot{\tilde{\omega }}=\dot{\omega }-{\dot{\omega }}_{d}\\ =-{\left(J+J_{VS}\right)}^{-1}\Omega (J+J_{VS})\omega +{\left(J+J_{VS}\right)}^{-1}T+d(t)-{\dot{\omega }}_{d} \end{array}$$

To deal with the disturbance of the inner loop, make the inner loop stable and the overall system track the given attitude angles, combined with adaptive technology, the following dynamic surface controller with variable-structure parameter is designed:(45)$$\begin{array}{l} T=-(J+J_{VS})[\Lambda \tilde{\omega }-{\left(J+J_{VS}\right)}^{-1}\Omega (J+J_{VS})\omega \\ +R^{T}\tilde{\theta }-{\dot{\omega }}_{d}+\hat{d}] \end{array}$$
where the last item is the estimation of disturbance, Λ = *diag*{*λ*_1_, *λ*_2_, *λ*_3_}(*λ_j_* > 0, *j* = 1, 2, 3) is the diagonal gain matrix to be designed, and the adaptive law of disturbance estimator is designed as
(46)$$\dot{\hat{d}}=(\Gamma +\Gamma^{*}(J_{VS}))(\tilde{\omega }-\hat{d})$$
where Γ and Γ^*^(*J_VS_*) satisfy:(47)$$\left\{ \begin{array}{l} \Gamma =diag\{\Gamma_{1},\;\Gamma_{2},\;\Gamma_{3}\}\\ \Gamma^{*}(J_{VS})=diag\{\Gamma_{1}{}^{*}(||J_{VS}||),\Gamma_{2}{}^{*}(||J_{VS}||),\Gamma_{3}{}^{*}(||J_{VS}||)\}\\ \Gamma_{j},\Gamma_{j}^{*}(J_{VS})>0,\;j=1,2,3 \end{array}\right.$$Γ is the adaptive gain matrix for the fixed structure flight mode, and Γ^*^(*J_VS_*) is the harmonic function matrix used to consider the variable structure mode. Equation (48) can be obtained by substituting the dynamic surface controller (45) into the angular rate error Equation (44) as follows:(48)$$\dot{\tilde{\omega }}=-\Lambda \tilde{\omega }-R^{T}\tilde{\theta }+\tilde{d}$$
where
(49)$$\tilde{d}=d-\hat{d}$$

Moreover, (49) expresses the estimation error of the disturbance, and its derivative is defined as
(50)$$\dot{\tilde{d}}=\dot{d}-\dot{\hat{d}}$$

**Theorem** **3.**
*Let*

(51)
$${\tilde{y}}_{d}={\left[\begin{array}{cc} {\tilde{\theta }}^{T} & {\tilde{\omega }}^{T} \end{array}\right]}^{T}$$

*For the HFV attitude nonlinear system, the dynamic surface controller (45) can stabilize the system, and the overall output error (51) is uniformly bounded.*


**Proof of Theorem** **3.**Define the Lyapunov function as follows:(52)$$\begin{array}{l} V=V_{1}+V_{2}\\ =\frac{1}{2}{\tilde{\theta }}^{T}\tilde{\theta }+\frac{1}{2}{\bar{\mu }}^{T}\bar{\mu }+\frac{1}{2}{\tilde{\omega }}^{T}\tilde{\omega }+\frac{1}{2}{\tilde{d}}^{T}{\left(\Gamma +\Gamma^{*}(J_{VS})\right)}^{-1}\tilde{d} \end{array}$$
(53)$$V_{1}=\frac{1}{2}{\tilde{\theta }}^{T}\tilde{\theta }+\frac{1}{2}{\bar{\mu }}^{T}\bar{\mu }$$
(54)$$V_{2}=\frac{1}{2}{\tilde{\omega }}^{T}\tilde{\omega }+\frac{1}{2}{\tilde{d}}^{T}{\left(\Gamma +\Gamma^{*}(J_{VS})\right)}^{-1}\tilde{d}$$

Let *η*_1_ = 1 and calculate the derivative of the first positive function of *V*_1_ as
(55)$$\begin{array}{l} {\tilde{\theta }}^{T}\dot{\tilde{\theta }}={\tilde{\theta }}^{T}[-(K_{1}+K_{1}^{*}(J_{VS}))\left|\tilde{\theta }\right|\mathrm{sgn}(\tilde{\theta })-(K_{2}+K_{2}^{*}(J_{VS})){\left|\tilde{\theta }\right|}^{\eta_{2}}\mathrm{sgn}(\tilde{\theta })\\ -(\varepsilon_{1}+\varepsilon_{1}^{*}(||J_{VS}||))S-(\varepsilon_{2}+\varepsilon_{2}^{*}(||J_{VS}||)){\left|S\right|}^{\eta_{3}}\mathrm{sgn}(S)+{\dot{\tilde{f}}}_{\theta }+R\tilde{\omega }+R\bar{\mu }]\\ \leq -{\tilde{\theta }}^{T}(K_{1}+K_{1}^{*}(J_{VS}))\tilde{\theta }-(K_{2}+K_{2}^{*}(J_{VS})){\left\|\tilde{\theta }\right\|}^{\frac{\eta_{2}+1}{2}}-{\tilde{\theta }}^{T}(\varepsilon_{1}+\varepsilon_{1}^{*}(||J_{VS}||))S\\ -{\tilde{\theta }}^{T}(\varepsilon_{2}+\varepsilon_{2}^{*}(||J_{VS}||)){\left|S\right|}^{\eta_{3}}\mathrm{sgn}(S)+{\tilde{\theta }}^{T}{\dot{\tilde{f}}}_{\theta }+{\tilde{\theta }}^{T}R\tilde{\omega }+{\tilde{\theta }}^{T}R\bar{\mu } \end{array}$$

According to Theorem 2, the sliding mode surface *S* converges to a bounded interval, namely:(56)$$\underset{t\to \infty }{\lim}{\left\|S\right\|}^{2}\leq \varpi$$
where
(57)$$\varpi =\min\left\{\left(\frac{\kappa }{\left(\varepsilon_{1}+\varepsilon_{1}^{*}(||J_{VS}||))\lambda_{\min}(\Lambda \right)}\right),{\left(\frac{\kappa }{\left(\varepsilon_{2}+\varepsilon_{2}^{*}(||J_{VS}||))\lambda_{\min}(\Lambda \right)}\right)}^{2/(\eta_{3}+1)}\right\}$$

Simultaneously, other functions of (54) meet the following conditions:(58)$$\begin{array}{l} \left|{\tilde{\theta }}^{T}(\varepsilon_{1}+\varepsilon_{1}^{*}(||J_{VS}||))S\right|\leq \frac{1}{2}{\tilde{\theta }}^{T}\tilde{\theta }+\frac{1}{2}{\left(\varepsilon_{1}+\varepsilon_{1}^{*}(||J_{VS}||)\right)}^{2}{\left\|S\right\|}^{2}\\ \leq \frac{1}{2}{\tilde{\theta }}^{T}\tilde{\theta }+\frac{1}{2}{\left(\varepsilon_{1}+\varepsilon_{1}^{*}(||J_{VS}||)\right)}^{2}\varpi \end{array}$$
(59)$$\begin{array}{l} \left|{\tilde{\theta }}^{T}(\varepsilon_{2}+\varepsilon_{2}^{*}(||J_{VS}||)){\left|S\right|}^{\eta_{3}}\mathrm{sgn}(S)\right|\leq \frac{1}{2}{\tilde{\theta }}^{T}\tilde{\theta }+\frac{1}{2}{\left(\varepsilon_{2}+\varepsilon_{2}^{*}(||J_{VS}||)\right)}^{2}{\left\|S\right\|}^{2\eta_{3}}\\ \leq \frac{1}{2}{\tilde{\theta }}^{T}\tilde{\theta }+\frac{1}{2}{\left(\varepsilon_{2}+\varepsilon_{2}^{*}(||J_{VS}||)\right)}^{2}\varpi^{\eta_{3}} \end{array}$$
(60)$${\tilde{\theta }}^{T}{\dot{\tilde{f}}}_{\theta }\leq \frac{1}{2}{\tilde{\theta }}^{T}\tilde{\theta }+\frac{1}{2}{\dot{\tilde{f}}}_{\theta }^{T}{\dot{\tilde{f}}}_{\theta }$$

Therefore, the (55) can be transformed into:(61)$$\begin{array}{l} {\tilde{\theta }}^{T}\dot{\tilde{\theta }}\leq -{\tilde{\theta }}^{T}[(K_{1}+K_{1}^{*}(J_{VS}))-\frac{3}{2}I-\frac{1}{2}RR^{T}]\tilde{\theta }-(K_{2}\\ +K_{2}^{*}(J_{VS})){\left\|\tilde{\theta }\right\|}^{\frac{\eta_{2}+1}{2}}+\frac{1}{2}{\dot{\tilde{f}}}_{\theta }^{T}{\dot{\tilde{f}}}_{\theta }+\frac{1}{2}{\bar{\mu }}^{T}\bar{\mu }+\frac{1}{2}(\varepsilon_{1}\\ +\varepsilon_{1}^{*}(||J_{VS}||){)}^{2}\varpi +\frac{1}{2}{\left(\varepsilon_{2}+\varepsilon_{2}^{*}(||J_{VS}||)\right)}^{2}\varpi^{\eta_{3}}+{\tilde{\theta }}^{T}R\tilde{\omega } \end{array}$$

The filter error satisfies the following conditions [32]:(62)$${\bar{\mu }}^{T}\dot{\bar{\mu }}\leq -\frac{1}{\tau }{\bar{\mu }}^{T}\bar{\mu }+\frac{1}{2}{\bar{\mu }}^{T}\bar{\mu }+\frac{1}{2}\nu^{T}\nu$$

Combining the above two formulas, the derivative of *V*_1_ is:(63)$$\begin{array}{l} {\dot{V}}_{1}\leq -{\tilde{\theta }}^{T}[(K_{1}+K_{1}^{*}(J_{VS}))-\frac{3}{2}I-\frac{1}{2}RR^{T}]\tilde{\theta }-(K_{2}\\ +K_{2}^{*}(J_{VS})){\left\|\tilde{\theta }\right\|}^{\frac{\eta_{2}+1}{2}}+\frac{1}{2}{\dot{\tilde{f}}}_{\theta }^{T}{\dot{\tilde{f}}}_{\theta }+\frac{1}{2}{\bar{\mu }}^{T}\bar{\mu }+\frac{1}{2}{\left(\varepsilon_{1}+\varepsilon_{1}^{*}(||J_{VS}||)\right)}^{2}\varpi \\ +\frac{1}{2}{\left(\varepsilon_{2}+\varepsilon_{2}^{*}(||J_{VS}||)\right)}^{2}\varpi^{\eta_{3}}+{\tilde{\theta }}^{T}R\tilde{\omega }-\frac{1}{\tau }{\bar{\mu }}^{T}\bar{\mu }+\frac{1}{2}{\bar{\mu }}^{T}\bar{\mu }+\frac{1}{2}\nu^{T}\nu \\ \leq -{\tilde{\theta }}^{T}[(K_{1}+K_{1}^{*}(J_{VS}))-\frac{3}{2}I-\frac{1}{2}RR^{T}]\tilde{\theta }-(K_{2}+K_{2}^{*}(J_{VS})){\left\|\tilde{\theta }\right\|}^{\frac{\eta_{2}+1}{2}}\\ -(\frac{1}{\tau }-1){\bar{\mu }}^{T}\bar{\mu }+\frac{1}{2}{\left(\varepsilon_{1}+\varepsilon_{1}^{*}(||J_{VS}||)\right)}^{2}\varpi +\frac{1}{2}{\left(\varepsilon_{2}+\varepsilon_{2}^{*}(||J_{VS}||)\right)}^{2}\varpi^{\eta_{3}}\\ +{\tilde{\theta }}^{T}R\tilde{\omega }+\frac{1}{2}\nu^{T}\nu +\frac{1}{2}{\dot{\tilde{f}}}_{\theta }^{T}{\dot{\tilde{f}}}_{\theta } \end{array}$$

The derivative of *V*_2_ in (52) can be obtained as
(64)$$\begin{array}{l} {\dot{V}}_{2}={\tilde{\omega }}^{T}(-\Lambda \tilde{\omega }-R^{T}\tilde{\theta }+\tilde{d})+{\tilde{d}}^{T}{\left(\Gamma +\Gamma^{*}(J_{VS})\right)}^{-1}(\dot{d}-\dot{\hat{d}})\\ =-{\tilde{\omega }}^{T}\Lambda \tilde{\omega }-{\tilde{\omega }}^{T}R^{T}\tilde{\theta }+{\tilde{d}}^{T}{\left(\Gamma +\Gamma^{*}(J_{VS})\right)}^{-1}\dot{d}+{\tilde{d}}^{T}\hat{d}\\ \leq -{\tilde{\omega }}^{T}\Lambda \tilde{\omega }-{\tilde{\omega }}^{T}R^{T}\tilde{\theta }-\frac{1}{2}{\tilde{d}}^{T}[I-{\left(\Gamma +\Gamma^{*}(J_{VS})\right)}^{-2}]\tilde{d}\\ +\frac{1}{2}{\left\|\dot{d}\right\|}^{2}+\frac{1}{2}{\left\|d\right\|}^{2} \end{array}$$

Combining the above two formulas, the derivative of *V* is:(65)$$\begin{array}{l} \dot{V}={\dot{V}}_{1}+{\dot{V}}_{2}\\ \leq -{\tilde{\theta }}^{T}[(K_{1}+K_{1}^{*}(J_{VS}))-\frac{3}{2}I-\frac{1}{2}RR^{T}]\tilde{\theta }-(K_{2}+K_{2}^{*}(J_{VS})){\left\|\tilde{\theta }\right\|}^{\frac{\eta_{2}+1}{2}}\\ -(\frac{1}{\tau }-1){\bar{\mu }}^{T}\bar{\mu }+\frac{1}{2}{\left(\varepsilon_{1}+\varepsilon_{1}^{*}(||J_{VS}||)\right)}^{2}\varpi +\frac{1}{2}{\left(\varepsilon_{2}+\varepsilon_{2}^{*}(||J_{VS}||)\right)}^{2}\varpi^{\eta_{3}}\\ +{\tilde{\theta }}^{T}R\tilde{\omega }+\frac{1}{2}\nu^{T}\nu +\frac{1}{2}{\dot{\tilde{f}}}_{\theta }^{T}{\dot{\tilde{f}}}_{\theta }-{\tilde{\omega }}^{T}\Lambda \tilde{\omega }-{\tilde{\omega }}^{T}R^{T}\tilde{\theta }-\frac{1}{2}{\tilde{d}}^{T}[I\\ -{\left(\Gamma +\Gamma^{*}(J_{VS})\right)}^{-2}]\tilde{d}+\frac{1}{2}{\left\|\dot{d}\right\|}^{2}+\frac{1}{2}{\left\|d\right\|}^{2}\\ =-{\tilde{\theta }}^{T}[(K_{1}+K_{1}^{*}(J_{VS}))-\frac{3}{2}I-\frac{1}{2}RR^{T}]\tilde{\theta }-(K_{2}+K_{2}^{*}(J_{VS})){\left\|\tilde{\theta }\right\|}^{\frac{\eta_{2}+1}{2}}\\ -(\frac{1}{\tau }-1){\bar{\mu }}^{T}\bar{\mu }-{\tilde{\omega }}^{T}\Lambda \tilde{\omega }-\frac{1}{2}{\tilde{d}}^{T}[I-{\left(\Gamma +\Gamma^{*}(J_{VS})\right)}^{-2}]\tilde{d}+C \end{array}$$
where
(66)$$\begin{array}{l} C=\frac{1}{2}{\left(\varepsilon_{1}+\varepsilon_{1}^{*}(||J_{VS}||)\right)}^{2}\varpi +\frac{1}{2}{\left(\varepsilon_{2}+\varepsilon_{2}^{*}(||J_{VS}||)\right)}^{2}\varpi^{\eta_{3}}+\frac{1}{2}\nu^{T}\nu \\ +\frac{1}{2}{\dot{\tilde{f}}}_{\theta }^{T}{\dot{\tilde{f}}}_{\theta }+\frac{1}{2}{\left\|\dot{d}\right\|}^{2}+\frac{1}{2}{\left\|d\right\|}^{2}-(K_{2}+K_{2}^{*}(J_{VS})){\left\|\tilde{\theta }\right\|}^{\frac{\eta_{2}+1}{2}} \end{array}$$

Equation (66) is a continuous bounded function, select the following appropriate parameters:(67)$$\lambda_{\min}(K_{1}+K_{1}^{*}(J_{VS}))>\frac{3}{2}+\frac{1}{2}\left\|R\right\|+\frac{a}{2}$$
(68)$$0<\tau <\frac{2}{a+2}$$
(69)$${\left(\Gamma +\Gamma^{*}(J_{VS})\right)}^{-2}-I>aI$$
(70)$$\Lambda >\frac{a}{2}I$$
where *a* is a positive constant, (65) can be rewritten as follows:(71)$$\dot{V}\leq -aV+C$$

According to the above formula, we can obtain:(72)$$0\leq V\leq [V(0)-\frac{C}{a}]e^{-at}+\frac{C}{a}$$
that is
(73)$$\underset{t\to \infty }{\lim}\max\{{\left\|\tilde{\theta }\right\|}^{2}+{\left\|\tilde{\omega }\right\|}^{2}\}\leq 2\underset{t\to \infty }{\lim}\max\{V\}\leq \frac{2C}{a}$$

Therefore, the output tracking error (51) satisfies the following inequality:(74)$$\underset{t\to \infty }{\lim}\sup\left\|{\tilde{y}}_{d}\right\|\leq \sqrt{\frac{2C}{a}}$$

If an appropriate parameter *a* is selected, the output tracking error can be arbitrarily small, as discussed in [32]. That is, the output error vector (51) is consistent and finally bounded, and the proof is complete. □

Therefore, the proposed adaptive sliding mode virtual FTC and dynamic surface robust control retain the reentry variable-structure HFV stable in the presence of sensor compound faults and disturbances, guaranteeing that the three attitude angles gradually track the reference command *θ_d_*.

**Remark** **4.**
*Because the variable structure has a size limit in real flight, J_VS_ is bounded. Therefore, regarding J_VS_ as an independent variable, any harmonic function or function element in the harmonic function matrix designed to adapt to a variable structure flight is bounded.*


**Remark** **5.**
*R will be affected by the output faults, but it is still measurable. If this nonlinear problem cannot be solved in the experiment, the following scheme can be used: by establishing an independent virtual model in software, the actual inputs are calculated to get the virtual outputs without faults; then, virtual outputs are imported to obtain a fault-free R. Finally, the sensor faults only affect the actual outputs but not R. The essence of this scheme is digital twin or approximation, with the same notation as R.*


## 4. Numerical Simulation Analysis

In this section, the residual/threshold curves of the designed fault detector, as well as the response curves of the attitude angles and angular rates under the fault-tolerant controller are shown using MATLAB to verify the effectiveness of the proposed FTC scheme.

### 4.1. Parameter Setting

Some key system and control parameters of the HFV are set first in the numerical simulations, where the fixed structure inertia matrix *J* is expressed as
(75)$$J=\left[\begin{array}{ccc} 554486 & 0 & -23002\\ 0 & 1136949 & 0\\ -23002 & 0 & 1376853 \end{array}\right]\mathrm{kg}\cdot m^{2}$$

According to the flight data of a variable-structure HFV with inertia (75), the variable structure inertia perturbation can be obtained as
(76)$$J_{VS}=\left\{ \begin{array}{l} \left[\begin{array}{ccc} 476t & 0 & -533t\\ 0 & 664t & 0\\ -718t & 0 & 3651t \end{array}\right]\mathrm{kg}\cdot m^{2},t_{\Delta 0}^{}\leq t<t_{\Delta 0}^{}+2\;s\\ \left[\begin{array}{ccc} 952 & 0 & -1066\\ 0 & 1328 & 0\\ -1436 & 0 & 7302 \end{array}\right]\mathrm{kg}\cdot m^{2},t\geq t_{\Delta 0}^{}+2\;s \end{array}\right.$$
where *t*_Δ0_ is the starting time of the HFV to change its fuselage structure. In this experiment, we set *t*_Δ0_ = 30 s.

The initial flight status is:(77)$$\left\{ \begin{array}{l} \theta_{0}={\left[\phi ,\alpha ,\beta \right]}^{T}={\left[0,0,0\right]}^{T}\mathrm{rad}\\ \omega ={\left[p,q,r\right]}^{T}={\left[0,0,0\right]}^{T}\mathrm{rad}/s \end{array}\right.$$

The angle reference command *θ_d_* is set to [0.6, 1, 0]*^T^* rad. Considering the electromagnetic environment of the battlefield, the disturbance is set as
(78)$$d(t)={\left[0.006\sin t,\;0.007\cos t,\;0.009\sin t\right]}^{T}$$

The compound faults of sensors at the three attitude angles are set as different time-varying faults occurring at the same time at *t* = 20 s. Thus,
(79)$$f_{\theta }(t)=\left\{ \begin{array}{ll} 0 & 0<t<20\;s\\ {\left[\cos(t-20),2\sin(t-20),3sin(t-20)\right]}^{T} & t\geq 20\;s \end{array}\right.$$

Gains of the nonlinear fault detector are selected as *α*^*^ = 6 and *β*^*^ = 11. Parameters of the adaptive sliding mode FTC system of the outer loop are as follows:$$\left\{ \begin{array}{l} K_{1}=diag\{0.004,0.005,0.007\}\\ K_{2}=diag\{0.005,0.007,0.009\} \end{array}\right.,$$
$$\left\{ \begin{array}{l} \varepsilon_{1}=8\\ \varepsilon_{2}=7 \end{array}\right.,$$
φ=diag{300,400,550}.

The parameters of the inner loop dynamic surface controller are:$$\left\{ \begin{array}{l} \Lambda =diag\{60,70,56\}\\ \Gamma =diag\{600,700,800\} \end{array}\right..$$

Finally, the variable structure reconciliation functions are set as:(80)$$\left\{ \begin{array}{l} K_{1}{}^{*}(J_{VS})=diag\{0.00003\sqrt{\left||J_{VS}|\right|},0.00002||J_{VS}||,0.0001||J_{VS}|{|}^{2}\}\\ K_{2}{}^{*}(J_{VS})=diag\{0.00003\sqrt{\left||J_{VS}|\right|},0.0002||J_{VS}||,0.00001||J_{VS}|{|}^{2}\} \end{array}\right.$$
(81)$$\left\{ \begin{array}{l} \varepsilon_{1}^{*}(||J_{VS}||)=0.0002\sqrt{\left||J_{VS}|\right|}\\ \varepsilon_{2}^{*}(||J_{VS}||)=0.0001\sqrt{\left||J_{VS}|\right|} \end{array}\right.$$
(82)$$\Gamma^{*}(J_{VS})=diag\{0.01\sqrt{\left||J_{VS}|\right|},0.01||J_{VS}||,0.008||J_{VS}|{|}^{2}\}$$

The system parameters are obtained using a real variable-structure HFV, and the control parameters are obtained using repeated experiments and optimizations. The ultimate goal is to guarantee that HFV reentry attitude system can track the desired output stably, quickly, and accurately.

### 4.2. Verification and Analysis

Figure 2 presents the threshold and residual of the proposed fault detection scheme when the system does not have a fault. The detection residual does not exceed the threshold while it fluctuates owing to disturbances though; thus, no fault occurs. The variable structure at *t* = 30 s fails to make the residual exceed the threshold, indicating that the fault detector is robust. Therefore, the detector can focus on fault alarms, independent of the environment and flight mode.

The attitude-tracking performance of a normal flight is shown in Table 1. A nominal controller is available when no fault exists. This study focuses on the results of FTC, and nominal tracking curves are not illustrated separately to avoid redundancy.

Figure 3 demonstrates that when sensor faults occur at *t* = 20 s, detection residual of the nonlinear fault detector increases significantly, exceeding the set threshold, and faults are successfully detected. The variable structure process at *t* = 30 s does not return the residual back under the set threshold; thus, the integrated detection algorithm can adapt to different flight modes.

Figure 4 presents the response curves of angular rates under nominal variable structure controller and variable structure FTC after the occurrence of faults. Figure 5 illustrates the response curves of attitude angles under nominal variable structure controller and variable structure FTC.

Comparison of the red and blue curves in Figure 4 and Figure 5 proves that after sensor faults, the system with the nominal controller is unstable and the output attitudes are severely jittered. Thirty seconds after changing the structure, the simultaneous occurrence of faults and structural variations makes all outputs diverge, where its performance becomes worse than when only faults occur in the time range of 20–30 s; thus, the nominal controller cannot be used. However, the FTC method can re-stabilize the system, as indicated by the solid red curves in Figure 4 and Figure 5.

In addition, a comparative analysis in Table 1 also verifies the superiority of the proposed FTC. Based on state-of-the-art FTC in reference [17], *e_st_*_2_ has six values, all of which are the values at *t* = 50 s. These values are meaningless and only indicate the divergence speed of the corresponding curve. ’—’ means that the response time cannot be obtained because of the curve divergence. Moreover, the units of all values are consistent with those in Figure 4 and Figure 5.

The existing algorithms in the above two comparisons have the application background of hypersonic vehicles. Figure 4 and Figure 5 and Table 1 show that with the compensation effect of the fault estimation algorithm in the virtual sliding mode controller (15) combined with the anti-disturbance controller (45), the proposed FTC scheme maintains the robust stability of the system. Each attitude angle and angular rate accurately tracked the reference command. Therefore, the proposed FTC with an integrated detection can manage coupled sensor compound faults and guarantee a reasonable flight performance for variable-structure HFVs.

## 5. Conclusions

This paper studies the fault detection and FTC problems of the reentry HFV nonlinear system with sensor compound faults and variable structure perturbation. Through the designed nonlinear fault detection observer, the fault can be detected by the residual and threshold values when a fault occurs at any attitude angle loop. Using the designed adaptive sliding mode FTC algorithm in outer loop, the sensor faults can be compensated in real time, and combined with the dynamic surface robust controller in inner loop, the output attitude angles can re-track the reference signals under faults and disturbance. Introducing the weighted integration detection, indirect fault compensation functions, and variable structure harmonic functions into the controller simplifies the FTC scheme design, because it does not require fault isolation and diagnostic observer channels, therefore, reducing the system complexity and calculation time. Furthermore, a comparative experiment based on MATLAB verifies the effectiveness of the designed scheme. The control technology proposed in this paper can create conditions for the highly reliable and ultra-maneuverable flight of the variable-structure HFV.

## Figures and Tables

**Figure 1 sensors-22-01523-f001:**
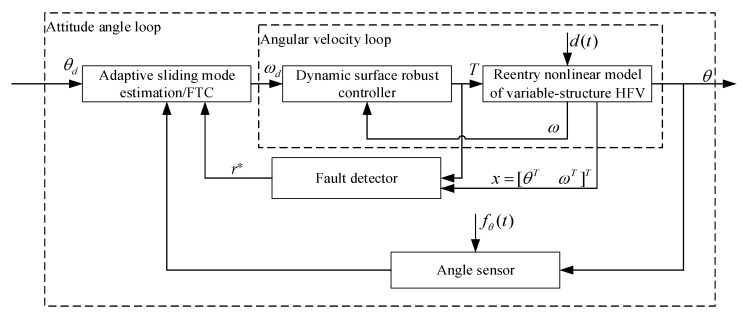
Block diagram of FTC for sensor faults in reentry attitude system.

**Figure 2 sensors-22-01523-f002:**
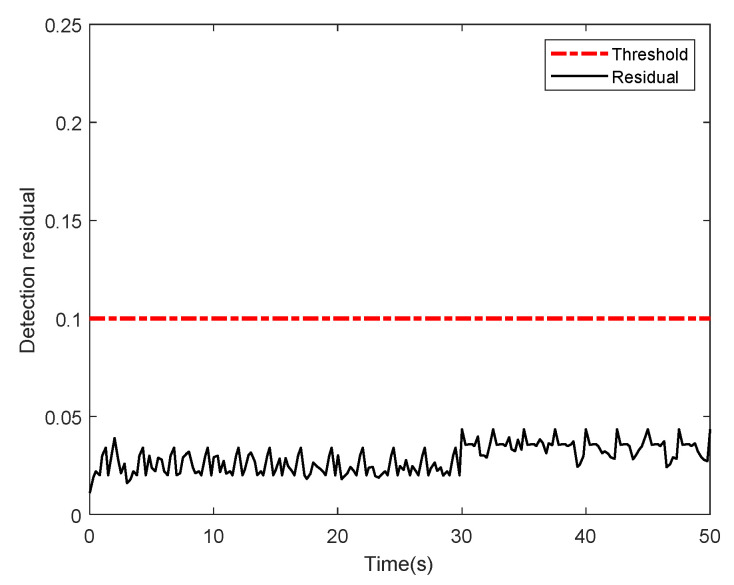
Detecting residual and threshold without fault.

**Figure 3 sensors-22-01523-f003:**
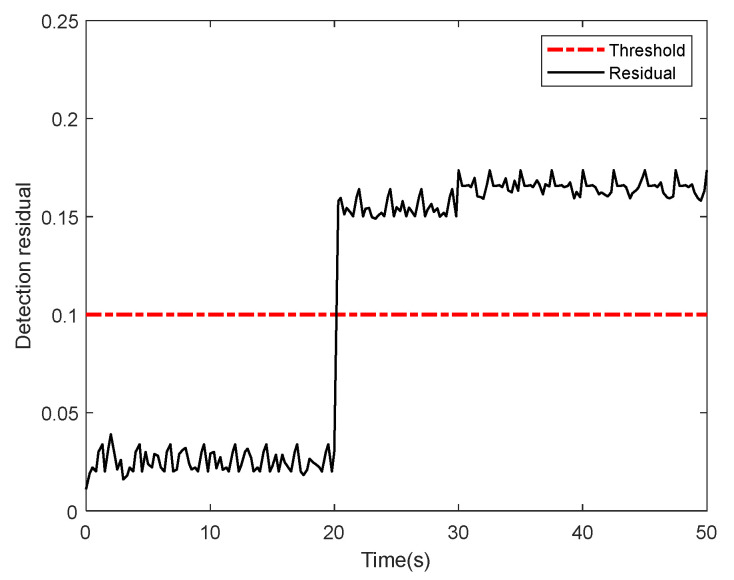
Detecting residual and threshold under sensor faults.

**Figure 4 sensors-22-01523-f004:**
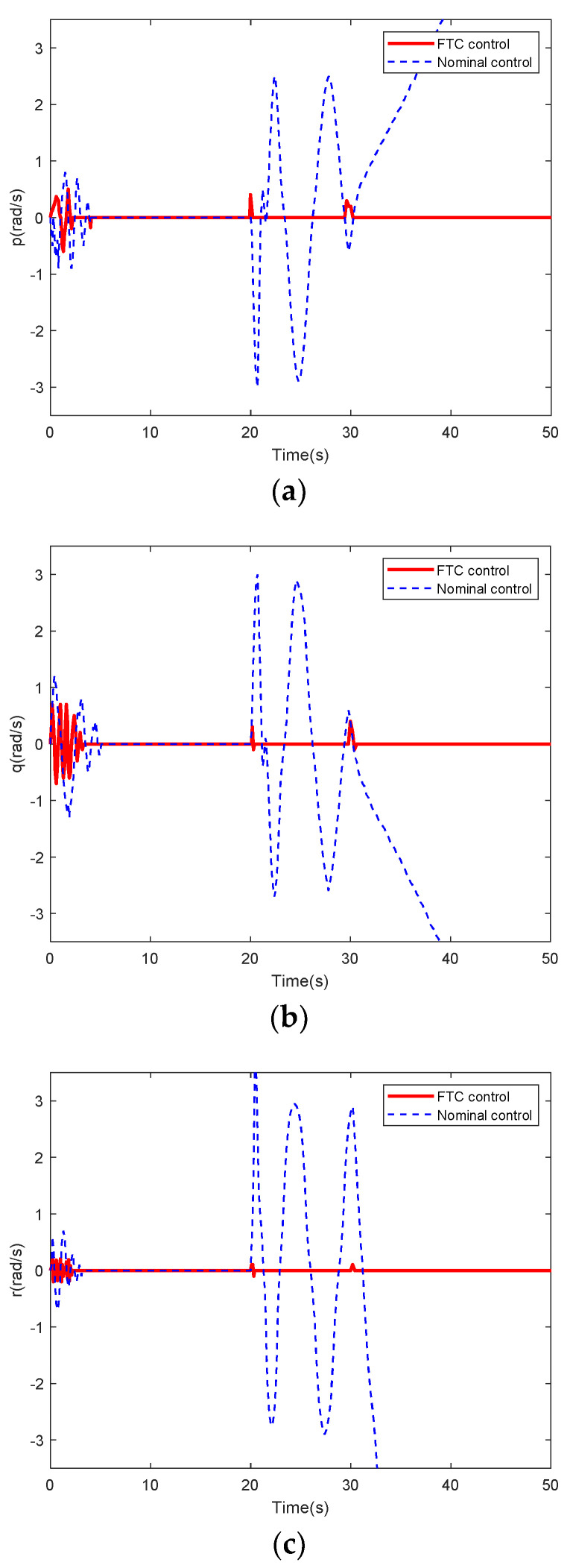
Response curves of three angular rate under sensor faults: (**a**) *p*; (**b**) *q*; and (**c**) *r*.

**Figure 5 sensors-22-01523-f005:**
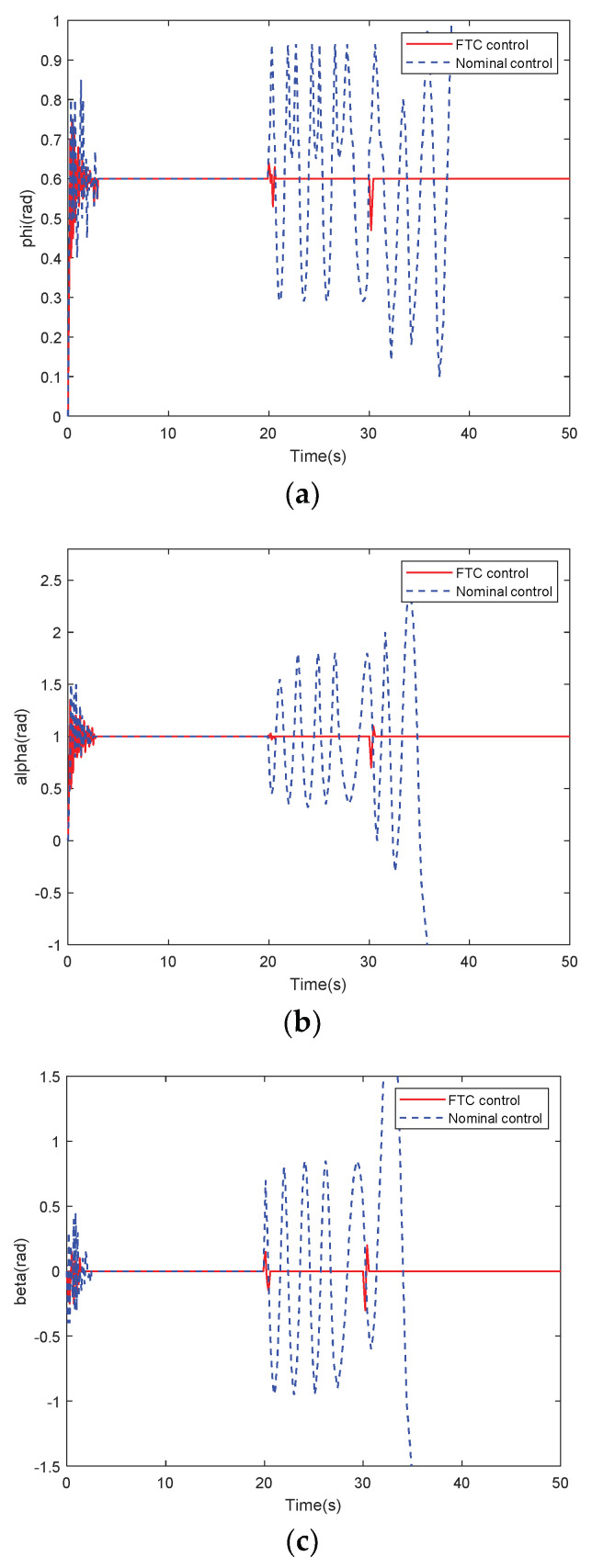
Response curves of three attitude angle under sensor faults: (**a**) *ϕ*; (**b**) *α*; and (**c**) *β*.

**Table 1 sensors-22-01523-t001:** Tracking performance comparison between state-of-the-art FTC in reference [17] and proposed FTC.

Indicator	*p*	*q*	*r*	*ϕ*	*α*	*β*	Controller
*t_r_* _1_	5.65	5.01	5.72	5.8	5.63	5.96	State of the art FTC
*e_st_* _1_	0.017	0.016	0.016	0.014	0.012	0.013	
*t_r_* _2_	-	-	-	-	-	-	
*e_st_* _2_	15.09	−11.24	−39.85	90.67	−57.61	69.44	
*t_r_* _2_	1.21	1.19	0.6	1.13	0.95	1.28	Proposed FTC
*e_st_* _2_	0.008	0.007	0.006	0.005	0.003	0.004	

## Data Availability

Not applicable.

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
