# Peer review of "Adaptive Sliding Mode Fault Compensation for Sensor Faults of Variable Structure Hypersonic Vehicle"

_sensors, 2022, doi:10.3390/s22041523_

Round 1
Reviewer 1 Report
This paper investigates the sensor fault detection and fault-tolerant control technology of a variable-structure hypersonic flight vehicle. The following points should be addressed:
1) Please explain how to calculate the derivative of \bar{\mu} in equation (42), as w_c seems not derivable
2)The writing should be improved to make the paper more readable.
3) The contributions should be declared more clearly.
4)ΔJ is assumed to be known ( in the simulation part )? If it is known,
there is no need to separate ΔJ from J in equation (1). If it is unknown,
the fault detection observer ( equation (11) ) can not be implemented.
5) Please declare clearly whether ω is measurable or not.
6)R is a function of θ. When the sensors of θ are at fault, the R calculated
by the θ measured is different from the R in equation (1), while they
are regarded as the same in this paper.
7) In theorem 2, it is stated that S converges to the equilibrium position,
however, it seems that only the boundedness of S is proved.
8) It is better to take ωd − ωc into consideration when designing ωc.
9) It is better to do some comparison with similar works in the literature.
Author Response
We thank reviewer 1 sincerely for the valuable comments and reviews on this study. We have spent a lot of time improving our works and revising the paper according to the reviewer 1’s comments. In the new revision, all modified parts are marked in red.
Please see the attachment.

Reviewer 2 Report
This paper presented “Adaptive sliding mode fault compensation for sensor faults of variable structure hypersonic vehicle”. This paper may have some significance. The authors do some simulations and get very beautiful results in this paper. However, this paper has several minor and critical problems as follows:
(1) The contribution of this paper is not convincing, which should be written in a more clear and remarkable ways. Moreover, the advantages of the fault compensation proposed in this paper should be clearly indicated in Introduction.
(2) Please, be sure to rearrange the Reference.
(3) In this paper, there are incorrect, including typos, grammar, spacing, punctuation, and commas. Please be sure to correct this part.
(4) The authors should explain in detail the reason for expressing the equation (2) as such.
(5) In equations (4) and (5), derivative values are bounded according to the assumption. They may have a strong assumption.
(6) In line 120, delta J is defined as the inertia perturbation caused by the active deformation of the fuselage. I know it is uncertainty. However, the authors had used it in sliding variable (equation (27)). The sliding variable should be defined as a prior knowledge in system. It means the sliding variable may not be converged.
(7) For fair comparison, it needs to be checked whether the compared algorithm fall into the insufficient relevance and must be mentioned in the revised paper.
Author Response
We thank reviewer 2 sincerely for the valuable comments and reviews on this study. We have spent a lot of time improving our works and revising the paper according to the reviewer 2’s comments. In the new revision, all modified parts are marked in red.
Please see the attachment.

Round 2
Reviewer 1 Report
All the problems have been well addressed in the revised paper.
Reviewer 2 Report
The authors responded appropriately to the reviewer's questions, and thus this paper can be sufficiently published.